# Post-Bariatric Hypoglycemia Is Associated with Endothelial Dysfunction and Increased Oxidative Stress

**DOI:** 10.3390/biomedicines10040916

**Published:** 2022-04-16

**Authors:** Roberta Lupoli, Ilenia Calcaterra, Giuseppe Annunziata, Giancarlo Tenore, Carmen Rainone, Luigi Schiavo, Brunella Capaldo, Matteo Nicola Dario Di Minno

**Affiliations:** 1Department of Molecular Medicine and Medical Biotechnology, Federico II University, 80131 Naples, Italy; 2Department of Clinical Medicine and Surgery, Federico II University, 80131 Naples, Italy; ileniacalcaterra@hotmail.it (I.C.); rainonecar@gmail.com (C.R.); bcapaldo@unina.it (B.C.); 3Department of Pharmacy, Federico II University, 80131 Naples, Italy; giuseppe.annunziata@unina.it (G.A.); giancarlo.tenore@unina.it (G.T.); 4Department of Medicine, Surgery, and Dentistry, University of Salerno, 84084 Salerno, Italy; lschiavo@unisa.it; 5Department of Translational Medical Sciences, Federico II University, 80131 Naples, Italy; matteo.diminno@unina.it

**Keywords:** bariatric surgery, hypoglycemia, endothelial function, oxidative stress

## Abstract

Post-bariatric hypoglycemia (PBH) is a potentially serious complication that may occur after bariatric surgery. Recurrent hypoglycemia may exert detrimental effects on vascular function. The aim of the present study was to evaluate endothelial function and oxygen reactive compounds in patients who experience PBH compared with controls. We performed a cross-sectional study on subjects with PBH (HYPO) and those without (NO-HYPO), detected by seven-day continuous glucose monitoring (CGM) performed at least twelve months after bariatric surgery. We enrolled 28 post-bariatric subjects (17.9% males, mean age 40.6 ± 10.7 years), with 18 in the HYPO group and 10 in the NO-HYPO group. In the two groups, we measured brachial artery flow-mediated dilation (FMD), oxidized low-density lipoproteins (oxLDL) and reactive oxygen metabolites (D-ROMs). The HYPO group had significantly lower FMD values than the NO-HYPO group (3.8% ± 3.0 vs. 10.5% ± 2.0, *p* < 0.001). A significant correlation was found between FMD and the time spent in hypoglycemia (rho = −0.648, *p* < 0.001), the number of hypoglycemic events (rho = −0.664, *p* < 0.001) and the mean glucose nadir (rho = 0.532, *p* = 0.004). The HYPO group showed significantly higher levels of D-ROMs (416.2 ± 88.7 UCARR vs. 305.5 ± 56.3 UCARR, *p* < 0.001) and oxLDLs (770.5 ± 49.7 µEq/L vs. 725.1 ± 51.6 µEq/L, *p* = 0.035) compared to the NO-HYPO group. In the multiple linear regression analysis, hypoglycemia independently predicted FMD values (β = −0.781, *p* < 0.001), D-ROMs (β = 0.548, *p* = 0.023) and oxLDL levels (β = 0.409, *p* = 0.031). PBH is associated with impaired endothelial function accompanied by increased oxidative stress.

## 1. Introduction

Post-bariatric hypoglycemia (PBH) is a potentially serious complication that may occur after Roux-en-Y gastric bypass (RYGB) or sleeve gastrectomy (SG)—the two most commonly performed bariatric procedures for the management of severe obesity [1]. The novel diagnostic tools available nowadays, which are able to detect even asymptomatic hypoglycemia, have shown a higher rate of PBH than commonly believed, involving up to 50% of surgical patients [2]. 

PBH can occur in both the postprandial and the fasting state, and may often present with disabling neuroglycopenic symptoms, greatly impairing quality of life [3].

The mechanisms underlying PBH are multifactorial and not yet completely understood. However, it is well-established that these episodes are partly the result of the exaggerated systemic appearance of ingested glucose, secondary to altered upper gastrointestinal anatomy due to surgical procedures. The brisk rise in blood glucose stimulates a rapid and excessive insulin secretion with subsequent hypoglycemia [3].

Generally, PBH can be controlled with adequate dietary modifications, but for patients with severe PBH, no approved medical therapy is currently available [3]. Furthermore, there is still no evidence as to whether repeated episodes of hypoglycemia, such as those occurring in PBH, could have clinical consequences and affect vascular function. 

In diabetic people, hypoglycemia is known to be associated with an increased risk of fatal and non-fatal cardiovascular events [4,5]. Hypoglycemia evokes a multifaceted response characterized by sympathoadrenal stimulation, proatherothrombotic and proinflammatory effects, which combine and increase cardiovascular risk [6]. It is also worth noting that even in healthy individuals, acute hypoglycemia impairs nitric oxide (NO)-mediated endothelial vasodilation, activates inflammatory processes, alters fibrinolytic balance and increases proatherothrombotic mechanisms [6]. Based on this knowledge, we hypothesized that in patients with PBH, the occurrence of repeated hypoglycemia may exert detrimental effects on vascular function. Thus, in the present study, we evaluated endothelial function, as measured by flow-mediated dilation (FMD) of the brachial artery, in bariatric patients who did or did not experience PBH. We also measured the plasma concentration of oxygen reactive compounds as a marker of endothelial activation and vascular damage.

## 2. Materials and Methods

### 2.1. Participants

We performed a cross-sectional study on non-diabetic subjects, aged 18–65 years, who had undergone RYGB or SG at least 1 year prior and had been referred to our clinic for follow-up visits and/or self-reported symptoms/signs of hypoglycemia under everyday life conditions. Exclusion criteria were: history of current alcohol or drug abuse; diseases or drugs interfering with glucose metabolism; history of diabetes; pregnancy; severe or unstable clinical conditions; revisional bariatric procedures; bariatric procedures other than RYGB or SG. The clinical evaluations of participants were conducted at the Department of Clinical Medicine and Surgery of Federico II University. All participants provided written, informed consent. The protocol was approved by the Federico II University Ethics Committee.

### 2.2. Continuous Glucose Monitoring (CGM) 

All participants underwent CGM for 7 days, with assessment of interstitial glucose (IG) levels every 5 min, 270 times per day (Dexcom G4 PLATINUM). The subcutaneous sensors were implanted at the Outpatient Diabetic Clinic and calibrated after two hours. Each participant was instructed to calibrate the CGM device twice daily or whenever alerted by the device. Glucose monitor data were downloaded using Dexcom StudioTM. Glucose variability (GV) was analyzed with regard to two principal components: amplitude and timing. The amplitude was expressed by coefficient of variation (CV), standard deviation of blood glucose (SD) and mean amplitude of glucose excursions (MAGE) calculated as the arithmetic mean difference between consecutive blood glucose peaks and nadirs (between the peaks) when differences were >1 SD of the mean glucose value. Moreover, CGM allowed us to derive: time spent in hypoglycemia (glucose levels < 54 mg/dL (3 mmol/L)) and the number of hypoglycemic events (glucose levels < 54 mg/dL (3 mmol/L) for more than 15 min) [7]. Detection of IG < 54 mg/dL for less than 15 min or ≥54 mg/dL were not considered hypoglycemic episodes. Participants experiencing at least one episode of hypoglycemia (defined as IG < 54 mg/dL for more than 15 min) during CGM were included in the HYPO group, whereas subjects showing no episode of hypoglycemia during CGM served as controls (NO-HYPO group). 

### 2.3. Laboratory Parameters

Plasma glucose concentrations were measured through the glucose oxidase method. 

Serum levels of oxidized low-density lipoproteins (oxLDL) and reactive oxygen metabolites (D-ROMs) were assessed as oxidative stress (OxS)-related biomarkers. D-ROM analysis was carried out by an automated analyzer (Free Carpe Diem, Diacron International, Grosseto, Italy) using relative commercial kits (Diacron International) according to the manufacturer’s instructions, as previously reported [8,9,10]. In detail, for the assessment of D-ROM levels, 10µL of serum was transferred into 1cm cuvettes containing 1 mL of R2 reagent (acetate buffer, pH4.8). The sample-containing mixture was gently mixed and 10 µL of R1 reagent (a chromogenic mixture consisting of aromatic alkyl-amine, A-NH2) was added. Cuvettes were mixed by inversion and samples were read at 546 nm (5 min, 37 °C) by an automated analyzer. D-ROMs are oxygen metabolites generated as a result of free radical attack; this occurs at the expense of biomolecules. As D-ROMs are more stable than other free radicals, they are adequately detectable and quantifiable. More specifically, the test used herein is based on the concept that, according to Fenton’s reaction, in the presence of iron, D-ROMs contained in serum generated alkoxyl (R−O*) and peroxyl (R−OO*) radicals which, in turn, oxidize an alkyl-substituted aromatic amine, producing a photometrically quantified [11,12,13], pink-colored derivative ([A−NH2*]+). D-ROMs are useful biomarkers of oxidative stress, determined according to the following ranges: (i) normal: 250–300 Units Carratelli (UCARR), (ii) borderline: 300–320 UCARR, (iii) mild oxidative stress: 321–340 UCARR, (iv) moderate oxidative stress: 341–400 UCARR, (v) high oxidative stress: 401–500 UCARR and (vi) very high oxidative stress: >500 UCARR, where 1 UCARR = 0.08 mg H_2_O_2_/dL [11,12,13].

Serum oxLDLs were measured using the LP-CHOLOX test (Free Carpe Diem, Diacron International, Grosseto, Italy). This test assesses a class of lipid peroxidation-deriving hydroperoxides derived from lipid peroxidation, mainly represented by oxidized cholesterol, which promotes the oxidation of ferrous iron (Fe^2+^) to ferric iron (Fe^3+^). The binding between the Fe^3+^ and the thiocyanate develops a colored complex that is spectrophotometrically measured at 505 nm. Briefly, 10 μL of serum was mixed with 1 mL of R1 reagent (indicator mixture) and two drops of R2 reagent (reduced iron). The solution was mixed by shaking, incubated at 37 °C for 2 min and centrifuged at 1400× *g* for 2 min. Supernatants were transferred into 1 cm cuvettes and read at 505 nm (37 °C). The absorbance values measured are directly proportional to the lipoperoxide concentrations, and the values are related to a standard solution (400 µEq/L). Results are expressed in µEq/L, and reference values are as follows: normal, ≤599 µEq/L; slight alteration, from 600 to 799 µEq/L; moderate alteration, from 800 to 999 µEq/L; strong alteration, ≥1000 Eq/L [14,15]. To avoid the potential confounding effect due to differences in LDL cholesterol, oxLDL levels were adjusted for LDL cholesterol serum levels [16]. 

### 2.4. Brachial Artery FMD

Endothelial function was evaluated by means of FMD—an accurate and non-invasive method that measures the dilation of the brachial artery in response to shear stress using high-resolution ultrasound [17]. 

Patients were asked to abstain from tobacco, caffeine and alcohol for at least 12 h before the examination. All study procedures were performed after an overnight fast in a temperature-controlled room (23 °C), after ≥10 min of rest in supine position (a small head pillow was accepted). Brachial artery FMD was evaluated by the same operator who was blinded to the participants’ clinical history. FMD was measured by ultrasound imaging, as described in the guidelines of the International Brachial Artery Reactivity Task Force [17] using an automatic edge-detection software (Cardiovascular Suite^®^, FMD studio, QUIPU Srl, Pisa, Italy). The examination consisted of measuring brachial artery diameter (BAD) at rest and after reactive hyperemia, induced by ischemia of the forearm. The measurement was made on a B-mode section of the artery, which was imaged above the antecubital fossa in the longitudinal plane by a linear ultrasound vascular transducer with a frequency of 10 MHz (Esaote^®^, MyLab 25 Gold, Pisa, Italy). Brachial artery diameter (BAD) at rest and flow velocity were recorded for 60 s. The blood pressure cuff was placed on the forearm 4–5 cm above the elbow joint line and inflated up to 70 mmHg above the individual’s systolic blood pressure to induce a transitory ischemia for 300 s. After 5 min, the cuff was deflated, and the diameter of the brachial artery was recorded for 240 s after deflation. FMD was calculated as: (Max post-ischemic diameter—basal diameter)/basal diameter × 100); then, FMD was expressed as the percentage increase of brachial artery diameter compared with the baseline value. The area under the curve (AUC) of hyperemia was also calculated as AUC of the shear rate during reactive hyperemia as related to baseline share rate. The overall duration of the exam was about 10–15 min. 

### 2.5. Statistical Analysis

Statistical analysis was performed with the IBM SPSS 27 system (SPSS Inc., Chicago, IL, USA). Continuous data were expressed as mean ± standard deviation (SD). The t-test was performed to compare continuous variables for independent samples. The Mann−Whitney U-test was used to compare variables with a skewed non-Gaussian distribution. The χ^2^ test with Fisher’s exact test was used to compare categorical variables. The relationships between continuous variables were examined using Spearman’s correlation (rho). All results were expressed as two-tailed values, with *p* values < 0.05 being statistically significant. To assess potential sources of heterogeneity, a sensitivity analysis was performed, which stratified patients according to the presence of cardiovascular risk factors. To adjust for potential confounders, linear regression analyses (stepwise method) were implemented with FMD values and stress oxidative markers as dependent variables, and gender, age, current BMI, fasting blood glucose, prevalence of smoking habit, dyslipidemia, hypertension, residual obesity and presence of hypoglycemia as independent variables.

### 2.6. Sample Size Calculation

With regard to sample size calculation, with a predefined difference in FMD values between the HYPO group and the control group of at least 40%, 9 subjects for each arm were needed to achieve an 80% power with a 5% α error.

## 3. Results

### 3.1. Characteristics of the Study Population

Thirty-five patients were assessed for inclusion in the study. Seven subjects were excluded due to a history of diabetes (three subjects), a bariatric procedure other than RYGB and SG (three subjects) and a revision bariatric procedure (one subject). Eventually, a total of 28 post-bariatric subjects (17.9% males, mean age 40.6 ± 10.7 years) were enrolled in the study. The clinical and demographic characteristics of the study population are reported in Table 1. In the overall sample, presurgical BMI was 44.5 ± 6.6 Kg/m^2^, and Excess Weight Loss (EWL) was 79.4 ± 11.3%. At the time of enrollment in the study, BMI was 26.8 ± 3.7 Kg/m^2^. At least one cardiovascular risk factor was reported by 64.3% of the study sample. Eighteen subjects (64.3%) experienced at least one episode of hypoglycemia during the CGM recording (HYPO group), while 10 subjects (35.7%) had no episodes of hypoglycemia (NO-HYPO group). No significant difference in major clinical and demographic characteristics was observed between the two groups. 

### 3.2. CGM Data

During the 7-day CGM, the HYPO group showed lower mean IG and IG nadir than the controls (Table 2). Overall, the HYPO group reported 5.9 ± 5.7 hypoglycemic events during the recording period, with 3.5 ± 3.3% of monitoring time spent at IG levels < 54 mg/dL and 16.0 ± 10.1% of monitoring time spent at IG levels 54–70 mg/dL (Table 2). In addition, HYPO subjects showed a significantly higher CV than NO-HYPO subjects (26.5 ± 7.3 vs. 20.4 ± 6.1, *p* = 0.035, Table 2). A significant positive correlation was found between CV and the time spent at IG levels < 54 mg/dL (rho = 0.616, *p* < 0.001). 

### 3.3. Vascular Reactivity and Oxidation Markers

BAD at rest was similar in HYPO and NO-HYPO subjects (3.7 mm ± 0.6 vs. 3.2 mm ± 0.6, *p* = 0.082). In contrast, the HYPO group showed significantly lower FMD values (3.8% ± 3.0 vs. 10.5% ± 2.0, *p* < 0.001, Figure 1) and a lower AUClog (9.1 s^−1^ ± 1.3 vs. 9.9 s^−1^ ± 0.5, *p* = 0.031). The difference in FMD was confirmed both in subjects with nocturnal hypoglycemia (3.8% ± 3.8 vs. 10.5% ± 2.0, *p* = 0.001) and in those with post-prandial hypoglycemia (3.9% ± 2.4 vs. 10.5% ± 2.0, *p* < 0.001).

As shown in Figure 2a, FMD values showed a significant inverse correlation with % time spent at IG < 54 mg/dL (rho = −0.648, *p* < 0.001). FMD was significantly lower in the highest tertile of % time spent at IG < 54 mg/dL compared with the lowest one, whereas a non-statistically significant trend was observed between the middle and the lowest tertile (Figure 3). In addition, FMD response correlated with the number of hypoglycemic events (rho = −0.664, *p* < 0.001, Figure 2b) and the mean IG nadir (rho = 0.532, *p* = 0.004, Figure 2c), but not with CV (rho = −0.256, *p* = 0.188, Figure 2d).

As for oxidation markers, the HYPO group showed significantly higher levels of D-ROMs (416.2 UCARR ± 88.7 vs. 305.5 UCARR ± 56.3, *p* < 0.001, Figure 4a) and oxLDLs (770.5 µEq/L ± 49.7 vs. 725.1 µEq/L ± 51.6, *p* = 0.035, Figure 4b) compared to the NO-HYPO group. D-ROMs correlated with FMD (rho = −0.512, *p* = 0.005), % time spent at IG < 54 mg/dl (rho = 0.478, *p* = 0.010) and the number of hypoglycemic events (rho = 0.488, *p* = 0.008), but not with the IG nadir (rho = −0.226, *p* = 0.247). oxLDLs correlated with FMD (rho = −0.465, *p* = 0.013), % time spent at IG < 54 mg/dl (rho = 0.410, *p* = 0.030), the number of hypoglycemic events (rho = 0.424, *p* = 0.024) and the IG nadir (rho = −0.502, *p* = 0.007).

Stratifying the population according to the presence of cardiovascular risk factors, we confirmed a significant difference in FMD between the HYPO and the NO-HYPO group in both subjects without cardiovascular risk factors (3.3% ± 2.6 vs. 9.9% ± 0.4, *p* = 0.002) and in those with ≥ 1 cardiovascular risk factor (4.2% ± 3.2 vs. 10.7% ± 2.4, *p* < 0.001). In the multiple linear regression analysis, after adjustment for resting BAD, gender, age, current BMI, fasting blood glucose, prevalence of smoking habit, dyslipidemia, hypertension and residual obesity, hypoglycemia independently predicted FMD values (β = −0.781, *p* < 0.001), D-ROMs (β = 0.548, *p* = 0.023) and oxLDL levels (β = 0.409, *p* = 0.031).

## 4. Discussion

Our results demonstrate a significant impairment of endothelial function in patients with PBH, as shown by a lower FMD response and increased serum levels of biomarkers related to OxS. The association between hypoglycemia and endothelial dysfunction is further supported by the finding that FMD was inversely correlated with the number of hypoglycemic events and the time spent in hypoglycemia, and directly correlated with the severity of hypoglycemia, as expressed by the IG nadir. 

Our findings highlight the potential detrimental impact of hypoglycemia on vascular function, even in non-diabetic subjects. In this context, it is also interesting to recall previous studies in healthy subjects which demonstrated that acute hypoglycemia, induced by a 2-h hypoglycemic clamp, activated a rapid proinflammatory, proatherothrombotic and procoagulant response accompanied by reduced FMD [18]. It is important to note that in our patients, the reduction in FMD response was independent of other cardiovascular risk factors, suggesting that hypoglycemia plays a key role in the impairment of endothelial function in PBH patients. Since these patients are exposed to recurrent hypoglycemic episodes, we can presume that endothelial damage progresses quite rapidly [19]. Interestingly, the reduction in FMD values was more evident in subjects experiencing > 2.8% of time spent in hypoglycemia, underlining the importance of the exposure time to low glucose levels in altering endothelial function. 

In the present study, HYPO subjects also presented higher levels of D-ROMs and oxLDLs than NO-HYPO subjects, suggesting a link between hypoglycemia, OxS and endothelial dysfunction. In addition, the close correlation between oxidation biomarkers and FMD, together with the finding that hypoglycemia independently predicted oxLDL and D-ROM levels, reinforces the view that high OxS may be an important driver of endothelial dysfunction in patients with PBH. OxS is a key feature in atherogenesis, and reactive oxygen species (ROS) are involved in all stages of the disease, from endothelial dysfunction to plaque formation and rupture [20]. D-ROMs are considered a surrogate marker of ROS production and OxS [21] since they measure the hydroperoxide metabolite levels (chemical: R–O–O–H) of global organic compounds (mainly of lipids, but also glycosides, amino acids and proteins) in serum [22,23]. Moreover, in cohort studies, D-ROMs were independent predictors of cardiovascular morbidity and mortality [22,24,25]. The oxLDLs, derived from the ROS-induced oxidation of lipoproteins, contribute to the atherosclerotic process through different mechanisms, including the expression of adhesion molecules, the migration and proliferation of smooth muscle cells and the formation of foam cells [26,27,28]. Furthermore, oxLDLs produce an imbalanced activation of nitric oxide synthase (NOS) expressed constitutively by endothelial cells (eNOS), thus facilitating the activation of the inducible isoform of this enzyme (iNOS) [29,30]. This impairment in the modulation of the eNOS/iNOS machinery enhances inflammatory processes within the vascular wall and contributes to atherosclerosis progression [31]. Our results are in line with previous in vitro and in vivo studies documenting a rise in ROS production during hypoglycemia [32,33]. Indeed, glucose deprivation stimulates the production of mitochondrial ROS and AMP−kinase in cultured endothelial cells from the human umbilical vein [32]. Furthermore, recurrent insulin-induced hypoglycemia in diabetic rats leads to increased OxS in brain mitochondria [33]. Also of great interest is the study by Ceriello et al. which showed a significant reduction in FMD response in Type 1 diabetic patients after 2-h hypoglycemia, together with a significant increase in both oxidative (plasma nitrotyrosine and plasma 8-iso-PGF2a) and inflammatory (sICAM-1 and IL-6) markers [34]. These abnormalities were mitigated by the infusion of an antioxidant agent (Vitamin C, 30 mg/min) [34]. 

In the present study, glucose variability (expressed as CV%) was significantly higher in the HYPO group compared to the NO-HYPO group and positively correlated with the time spent in hypoglycemia, as shown in previous reports [35,36]. However, we could not find any significant correlation between CV and FMD, or with other measurements of glucose variability (SD, MAGE). Similar results have been reported by Peña et al. who found that hypoglycemia, but not glycemic variability, correlated with reduced endothelial function in children with Type 1 diabetes without cardiovascular risk factors [37]. The lack of a significant association between glucose variability and FMD in the present study could be at least in part explained by the fact that daily glucose fluctuations and hence, glucose variability, were only moderately elevated in our patients since they were not diabetic. 

To the best of our knowledge, this is the first study showing a relationship between hypoglycemia, endothelial dysfunction and oxidative stress in subjects with PBH. Our results might appear to conflict with the evidence that bariatric surgery is associated with an improvement in subclinical atherosclerosis, endothelial function [38] and oxidative stress [39]. In actuality, the present data do not question the benefits of bariatric surgery for cardiovascular risk, which is consistently confirmed in recent long-term clinical studies [40,41]; rather, they indicate that the occurrence of hypoglycemia, a common complication of bariatric surgery [42], could hinder these beneficial effects. 

These findings may have relevant clinical implications since endothelial dysfunction and oxidative stress are important drivers of the atherosclerotic process and predict cardiovascular events [22,24,25,43]. Of note, the difference in FMD between HYPO subjects and NO-HYPO subjects is clinically significant, considering that a 1% reduction in FMD is associated with a 9–13% increase in cardiovascular risk [44,45]. In the light of the evidence that PBH exerts potential adverse effects on the vascular system, it is important to implement appropriate therapeutic interventions to prevent and manage this complication [46,47]. In addition, a careful preoperative assessment should be performed to identify patients at a higher risk to develop PBH [48].

Some limitations of our study should be discussed. Most of the patients examined presented concomitant cardiovascular risk factors, which may potentially impact endothelial function. However, the subgroup analysis showed a significantly lower FMD in the HYPO group than in the no-HYPO group, regardless of the concomitant presence of cardiovascular risk factors. The role of hypoglycemia in the impairment of endothelial function was also reinforced by the results of the multivariate analysis after adjustment for confounders. Another factor potentially affecting the comparison of FMD between HYPO and no-HYPO subjects is a difference in resting BAD [49]; for this reason, in the present study we measured resting BAD, which was similar between the two groups, thus ruling out such a possibility. Furthermore, to reduce operator bias and optimize repeatability of the results, we used a software that enables the automated assessment of vessel diameter and flow velocity. Finally, the study sample is limited. Although we performed a sample size calculation to estimate the target enrollment number, this type of calculation for small-sized studies is somewhat artificial and requires caution [50]. However, our results show robust statistical significance and high internal consistency, making us confident about their generalizability and reproducibility. 

In conclusion, patients with PBH appear to have impaired endothelial function, likely related to increased oxidative stress, which is independent of cardiovascular risk factors. The impact of these abnormalities on long-term cardiovascular outcomes needs to be investigated.

## Figures and Tables

**Figure 1 biomedicines-10-00916-f001:**
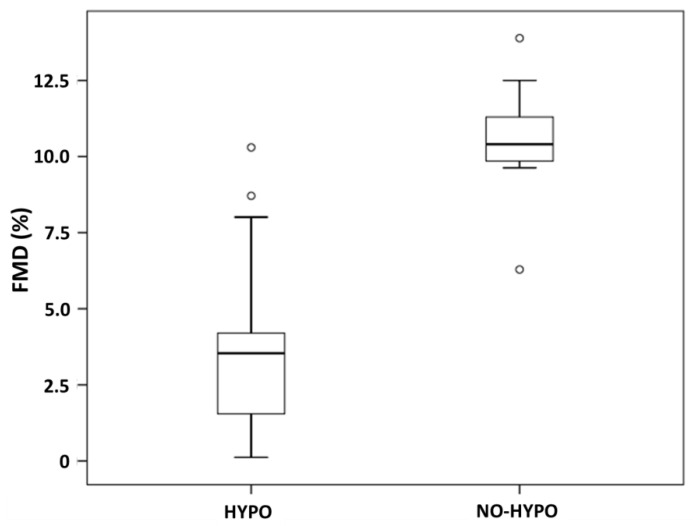
Flow-mediated dilation in HYPO and NO-HYPO groups.

**Figure 2 biomedicines-10-00916-f002:**
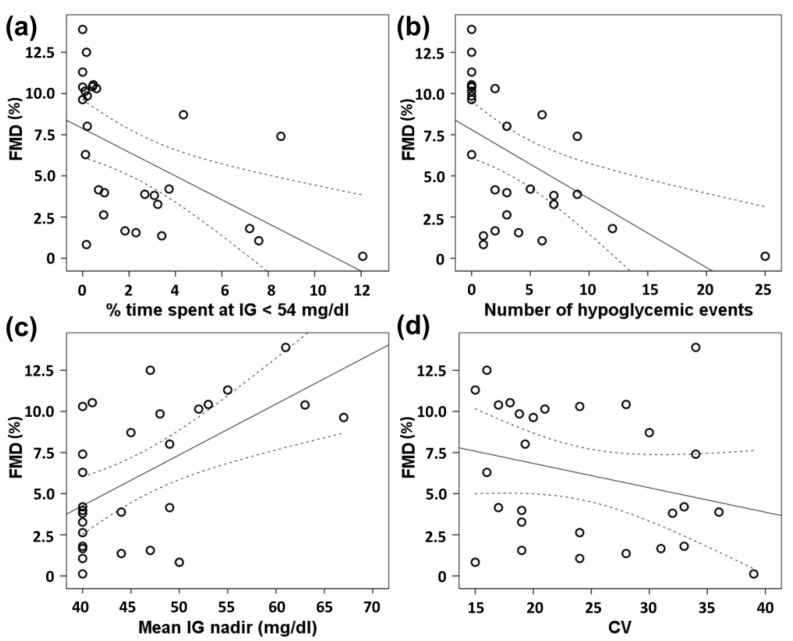
Scatter plot of Spearman correlations between flow-mediated dilation values (FMD) and: (**a**) % time spent at interstitial glucose (IG) < 54 mg/dL; (**b**) number of hypoglycemic events; (**c**) mean IG nadir; (**d**) coefficient of variation (CV).

**Figure 3 biomedicines-10-00916-f003:**
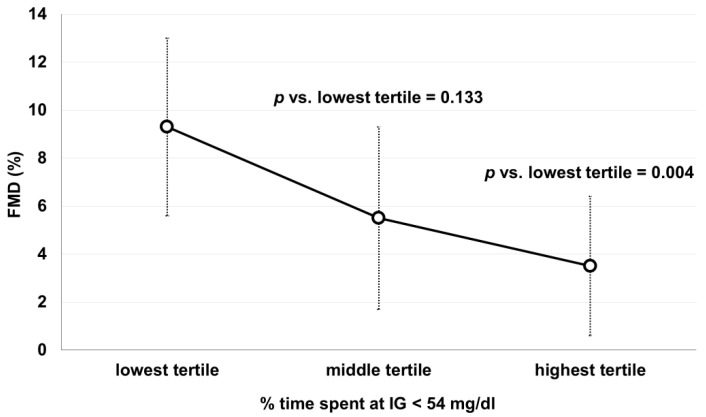
Flow-mediated dilation values (FMD) according to tertiles of % time spent in hypoglycemia (IG < 54 mg/dL). IG: interstitial glucose. Lowest tertile: time in hypoglycemia ≤ 0.2%; Middle tertile: time in hypoglycemia 0.3–2.8%; Highest tertile: time in hypoglycemia > 2.8%. Comparisons were made with Mann−Whitney U-test.

**Figure 4 biomedicines-10-00916-f004:**
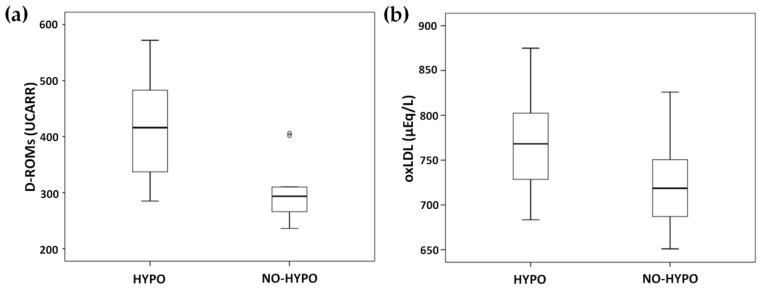
Oxidative stress-related biomarkers in HYPO and NO-HYPO groups: (**a**) D-ROMs; (**b**) oxLDL.

**Table 1 biomedicines-10-00916-t001:** Clinical and demographic characteristics of study population.

	Total *n* = 28	HYPO *n* = 18	NO-HYPO *n* = 10	*p* Value HYPO vs. NO-HYPO
Male gender—*n* (%)	5 (17.9)	3 (16.7)	2 (20)	1.000
Age (years)	41 ± 11	40 ± 12	41 ± 10	0.832
Current BMI (Kg/m^2^)	27 ± 4	27 ± 4	26 ± 3	0.296
Preoperative BMI (Kg/m^2^)	44 ± 7	45 ± 7	43 ± 6	0.377
TBWL (%)	39 ± 6	39 ± 5	39 ± 8	0.982
EWL (%)	79 ± 11	78 ± 12	83 ± 10	0.317
RYGB—*n* (%)	12 (42.9)	9 (50)	3 (30)	0.434
Fasting blood glucose (mg/dL)	74 ± 8	73 ± 9	75 ± 6	0.387
Smoking habit *n* (%)	10 (35.7)	7 (38.9)	3 (30)	0.703
Dyslipidemia *n*—(%)	2 (7.1)	0 (0)	2 (20)	0.119
Hypertension *n*—(%)	4 (14.3)	1 (5.6)	3 (30)	0.116
Obesity *n*—(%)	5 (17.9)	4 (22.2)	1 (10)	0.626
≥1 vascular risk factor—*n* (%)	18 (64.3)	11 (61.1)	7 (70)	0.703

BMI: body mass index; TBWL: total body weight loss; EWL: excess weight loss; RYGB: Roux-en-Y gastric bypass.

**Table 2 biomedicines-10-00916-t002:** Continuous glucose monitoring parameters.

	HYPO *n* = 18	NO-HYPO *n* = 10	*p* Value HYPO vs. NO-HYPO
Mean IG (mg/dL)	90 ± 8	104 ± 14	0.009 *
Mean IG peak (mg/dL)	195 ± 41	203 ± 58	0.832 *
Mean IG nadir (mg/dL)	43 ± 4	53 ± 9	0.001 *
CV (%)	26 ± 7	20 ± 6	0.031
SD (mg/dL)	24.0 ± 6.9	21.9 ± 9.5	0.506
MAGE (mg/dL)	69.8 ± 29.7	54.2 ± 15.4	0.155
% time spent at IG < 54 mg/dL	3.5 ± 3.3	0.15 ± 0.17	<0.001
% time spent at IG 54–70 mg/dL	16.0 ± 10.1	3.2 ± 3.5	<0.001
% time spent at IG 71–130 mg/dL	73.5 ± 13.9	85.9 ± 13.0	0.035
% time spent at IG 131–190 mg/dl	6.4 ± 5.0	9.4 ± 10.6	0.524 *
% time spent at IG > 190 mg/dL	0.5 ± 0.6	1.3 ± 3.3	0.689 *

IG: interstitial glucose; CV: coefficient of variation; SD: standard deviation of blood glucose; MAGE: mean amplitude of glucose excursions. * Mann−Whitney U-test was used for non-normally distributed variables.

## Data Availability

The data presented in this study are available on request from the corresponding author. The data are not publicly available due to ethical reasons.

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
