# Peer review of "Post-Bariatric Hypoglycemia Is Associated with Endothelial Dysfunction and Increased Oxidative Stress"

_biomedicines, 2022, doi:10.3390/biomedicines10040916_

Round 1

Reviewer 1 Report

Roberta Lupoli et al. performed an interesting study to investigate how hyperglycemia in post-bariatric patients influences endothelial cell function measured by FMD and measurement unfavorable for endothelial cells oxLDL and reactive oxygen species. Although the study raises a significant point regarding the propensity of bariatric patients to CVD, I have a few major and minor remarks.

MAJOR 

  1. The measurement of interstitial glucose levels every 5 min per day using a glucose sensor does not always indicate the correct glucose concentration and often lowers it, showing hypoglycemia, which is not detected in capillary blood at the same time using a glucometer. These are my personal observations of using glucose sensor Medtronic's ENLiTE and glucometer simultaneously. Your HYPO group spent in hypoglycemia (<54-70 mg/ml) about 20% of the time during seven days, NO-HYPO group only 3,5%. I think patient wearing glucose sensors should also have their glucometer to confirm hypoglycemia below 54 mg/dl.

  1. It is to small group of patients (HYPO n=18, NO-HYPO n=10) to create reliable Spearman correlation and multivariate analysis.

  1. Please mention in the method section the procedure in FMD measuring in patients with hypertension (4 patients from 28). Do they take the medication before FMD assessment or not? You mention only tobacco, caffeine, and alcohol before the examination.

MINOR

- Mechanism of hypoglycemia in PBP has to be precisely explained in the introduction or discussion. Why the glucose temporarily drops in some patients and not in others?

- Please explain the unit of measure UCAR in D-ROMs measurement.

- Please explain the abbreviations RYGB, SG OxS.

-Please provide a value of proper FMD in healthy people (7-10%).

- Clarify the BMI value in PBP obesity. In Tab. 1 average BMI is about 26-27, which indicates overweight not obesity (BMI 25-30 = overweight = first degree of obesity).

- There is no panel A in Fig. 1.

-Provide a unit for D-ROMs, oxLDL, FMD in Fig 1, 3 and 4.

- Provide a study limitation in the method section not in discussion.

Reviewer 2 Report

General comments

2.1. Participants

I suggest better justifying the sample size that was chosen so that it is sufficient to carry out statistical studies. Because this section mentions that several patients were excluded. But no figures are given to specify how many were excluded and why. A scheme can be made with the initial number of patients and each of those excluded and their causes. Thus, it can be understood how the calculated size of the sample was reduced (as was mentioned in the discussion) to respect the inclusion criteria.

Discussion

Although these studies were not done on patients, the proposal for the use of vitamin C is interesting. Therefore, the authors could indicate the dose suggested by the study they refer to

These abnormalities were mitigated by the infusion of an antioxidant agent (Vitamin C, 30 mg/min) [40].

These abnormalities were mitigated by the infusion of an antioxidant agent (Vitamin C) [40].

The study results are interesting; it only remains to highlight its usefulness and clinical application at the end. The physician reading this document should understand the importance of measuring postbariatric hypoglycemia and that PBH is associated with impaired endothelial function, accompanied by increased oxidative stress.

Figures 1 and 2.

The numbers that appear on the X and Y axes of the two graphs are small. If it is possible could be increased like the numbers in figure 3. Same comment for figure 4.

Round 2

Reviewer 1 Report

All my suggestions were made entirely. I'm afraid I only have to disagree with one explanation presented below.

I think the manuscript will be ready for publication if the editor so decides.

“In our case, significant correlations and regressions were obtained despite the limited study sample, suggesting that increasing the sample size would make the results even more statistically significant.”

I can't entirely agree with this statement. The large group of patients is connected with large diversity (Gaussian distribution) and does not always share the exact correlation obtained in a small homogenous group.